# Mindfully and confidently digital: A mixed methods study on personal resources to mitigate the dark side of digital working

**Elizabeth Marsh** [1‡]*, **Elvira Perez Vallejos**[2‡], **Alexa Spence**[1‡]

**1** Department of Psychology, University of Nottingham, Nottingham, United Kingdom, **2** School of Medicine (Nottingham Biomedical Research Centre) and School of Computer Science, University of Nottingham, Nottingham, United Kingdom

‡ EM was the principal author on this work. EPV and AS were co-authors on this work.
* elizabeth.marsh@nottingham.ac.uk

**Data Availability Statement:** The dataset has now been deposited on the UK Data Service's Reshare site: https://reshare.ukdataservice.ac.uk/856732.

## Abstract

A growing body of research demonstrates the potential of mindfulness to reduce employee stress. However, with work increasingly migrating from the physical to the digital workplace, evidence is lacking on how mindfulness might help employees live healthy digital working lives. In addition, employees' confidence when using the digital workplace is seen as important for productivity but may also play a role in reducing well-being impacts from digital working. Using the Job-Demands Resources model as a theoretical foundation, 142 workers were surveyed regarding their levels of trait mindfulness and digital workplace confidence, along with their experiences of the dark side effects (stress, overload, anxiety, Fear of Missing Out and addiction) and well-being outcomes (burnout and health). 14 workers were also interviewed to provide qualitative insights on these constructs. Results from regression analyses indicated that more digitally confident workers were less likely to experience digital workplace anxiety, while those with higher mindfulness were better protected against all of the dark side of digital working effects. Interview data indicated ways in which digital mindfulness helps protect well-being, as well as how digital workplace confidence enables healthier digital habits.

## Introduction

The digital workplace is recognized as a strategic asset for organisations [1] and a central feature of modern working life post-pandemic [2]. Yet just as employees encounter risks in the physical workplace, they may experience dark side of digital working effects such as stress, overload, anxiety and addiction that can have damaging consequences for employee well-being [3, 4]. While organisational and technological solutions are critical to addressing these risks, workers also need to be equipped with a range of skills and tactics to work effectively in the digital workplace [5]; for instance, being more mindful and more confident when utilising workplace technologies [6, 7].

**Funding:** The authors (EM, AS, EPV) acknowledge the support by the Economic and Social Research Council (https://www.ukri.org) grant number: ES/P000711/1. AS and EPVacknowledge the support of theUK Research and Innovation (UKRI) Trustworthy Autonomous Systems Hub (https://tas.ac.uk) EPSRC project ref.EP/V00784X/1 and Horizon (https://www.ukri.org) EPSRC project ref. EP/TO22493/1. EPV acknowledges the support of the NIHR Biomedical Research Centre (https://nottinghambrc.nihr.ac.uk). The funders had no role in study design, data collection and analysis, decision to publish, or preparation of the manuscript.

**Competing interests:** The authors have declared that no competing interests exist.

Mindfulness has shown potential as a personal resource to reduce employee stress and burnout and protect worker health [8–10] but is as yet underexplored as a protective factor in the digital workplace; while findings on digital workplace confidence have been somewhat mixed and warrant further investigation [5]. The current study also responds to the need for more qualitative and mixed method studies in the dark side of digital working literature [11] as well as the need for more studies grounded in theory [5].

## Job Demands-Resources theory

Job Demands-Resources theory (JD-R) [12, 13], integrates both positive (resources) and negative (demands) characteristics of jobs and these can be physical, psychological, social or organisational. High job demands can result in technostress and overload, and combined with a lack of resources can lead to burnout and health problems [14]. Personal resources refer to individual attributes related to core self-concept that help people cope with difficulties and adapt to circumstances [15]. Like job resources, they can moderate between job demands and negative well-being outcomes [15]. Both mindfulness and self-efficacy related to technology [16] have been indicated as personal resources that can help to reduce the impact of job demands on employee well-being, thereby acting as protective factors. The JD-R model has strong empirical support and is one of the most popular and widely used within the organisational stress literature [17] enabling a wide range of demands, resources and outcomes to be studied in a variety of different contexts [14].

## Digital workplace dark side effects

The present study assessed workers' experiences of dark side effects as they relate to the broad suite of digital workplace tools (e.g., e-mail, mobile devices, instant messaging). It focused on perceived rather than objective digital workplace demands in order to explore the psychological experience of working digitally [5]. Five dark side effects were included: digital workplace stress, overload, anxiety, Fear of Missing Out, and addiction (see **S1 Appendix** for construct definitions). We propose that these digital workplace dark side effects will be positively associated with employee burnout but negatively associated with health, trait mindfulness and digital workplace confidence.

Digital workplace technologies such as e-mail, instant messaging and mobile devices have been shown to contribute to perceptions of stress by employees [18, 19]. Employees may experience stress in relation to technology (technostress)when they struggle to adapt to the evolving digital workplace environment [20] or cope with its demands [21]. It has been found to lead to cognitive and affective outcomes such as higher burnout and lower job satisfaction [3, 22].

In a study of knowledge workers, Karr-Wisniewski and Lu [23] identified three aspects of digital workplace overload, relating to the amount of information and communication in the digital workplace as well as the number of features of digital workplace applications. Overload can involve a sense of having to work harder and faster [20] resulting in affective strain for digitally overloaded workers [24].

Anxieties about using digital workplace devices and applications have been found to contribute to a sense of technology-related strain for employees [25] and, along with technology-related stress, can be detrimental to mental well-being [26]. In a review of two decades of research on computer anxiety, Powell [27] identifies that it can be understood as a correlate of computer self-efficacy, with more confident users less likely to experience computer anxiety.

Anxiety may also appear in the form of Fear of Missing Out (FoMO) on both information and relationships, potentially leading to diminished work and career opportunities [28]. Although research is, as yet, sparse in this area, evidence suggests that when employees

experience FoMO in the digital workplace it can contribute to problematic IT use [29] and perceived work overload [30]. Fridchay & Reizer [31] found that FoMO led to higher levels of job burnout for workers as well as negatively impacting performance (both directly and indirectly via burnout).

Employees may find themselves using the digital workplace in an excessive and compulsive manner. While not formally defined as an addiction, maladaptive use of digital workplace tools such as e-mail and smartphones has been found to lead to elevated stress and burnout, poorer health and well-being, and lower satisfaction and productivity for employees [32, 33]. Indeed, Salanova et al. [34] argue that technoaddiction is the most important way in which technostress is experience, along with technostrain. Despite these detrimental outcomes, digital workplace addiction has received relatively little attention in the dark side literature [5].

## Employee well-being

Burnout is characterised as a state of exhaustion and fatigue [35]. While job demands can contribute to burnout, job and personal resources can protect against it [36]. Within the dark side of digital working literature, burnout is the most widely studied negative well-being outcome, and although its ultimate manifestation in the form of potential health impacts is less well understood [5] burnout is known to be detrimental to health [37]. There is evidence of associations between burnout and technology-related stress, as well as with overload, addiction and FoMO [28, 38, 39]. Maier et al. [22] found that technology-related stress experienced by employees contributed to techno-exhaustion and ultimately to work-exhaustion. Based on the above evidence, burnout may have the potential to mediate between dark side of digital working effects and health outcomes.

Good health is thought to comprise physical, mental and social well-being [40]. Although work can be a key component of good health [41], job demands can be detrimental to it [17]. The demands of workplace technology and resulting stress can negatively impact health [42, 43], however evidence of potential impacts of the digital workplace on employee health is not yet extensive [5].

## Personal resources

**Mindfulness.**   Defined simply, mindfulness is a state of consciousness that involves paying attention in the present moment intentionally and non-judgmentally [44–46]. It is contrasted with autopilot which is characterised as being unaware of one's actions and external events, potentially leading to perpetuation of undesirable thoughts and behaviours [44].

This study uses the secular approach to mindfulness derived from Buddhism [47] to focus on the role that attention and awareness may play in mitigating the impact of digital workplace demands on employee well-being. It focuses therefore on the broad, dispositional construct of trait mindfulness, rather than an IT-specific secondary trait as is articulated in the cognitive psychological stream of mindfulness research [48, 49].

In a review of over 55 years of mindfulness research, Baminiwatta and Solangaarachchi [9], demonstrate that evidence of the mental and physical health benefits of mindfulness is now considered extensive. This is particularly the case in the workplace where it has shown potential to reduce stress, anxiety and burnout, promote mental health and well-being, and improve work engagement [50–55]. Early findings also suggest mindfulness may help reduce stress in relation to the digital workplace in such a way that it is protective of employee well-being [56–58]. Pflügner and Maier [6] found that more mindful employees exhibited lower levels of aspects of technostress including overload, complexity, invasion and uncertainty, arguing that this may have been due to changes in cognitive and emotional processes due to mindfulness.

In context of J-DR model, mindfulness has been suggested as a personal resource that could be protective of well-being by buffering the impact of job demands. In this context it may work by reducing the negative stress appraisal process that leads to burnout [59] and influencing perceptions of job demands through its effect on how attentional resources are used [60]. In a systematic review and meta-analysis of workplace mindfulness training randomized control trials, Bartlett et al. [8] identify the potential of integrating mindfulness within the Job Demands-Resources theory, and call for further work exploring mindfulness as a personal resource.

Based on this research evidence, trait mindfulness (TM) is here explored as a personal resource that may reduce negative well-being impacts of digital workplace dark side effects by acting as either a moderator between them, or as an antecedent of the dark side effects.

**Digital workplace confidence.** Computer self-efficacy is defined as 'an individual's ability to apply his or her computer skills to a wider range of computer related tasks.' [61]. It is a personality trait that is both dynamic and specific to the technology context [62]. There is evidence to suggest that computer self-efficacy can help to reduce the stressful aspects of digital workplace technology use [7, 63, 64], thereby helping to protect well-being in the digital workplace. In a qualitative synthesis of 105 technostress studies, La Torre et al. [11] found that computer self-efficacy might mitigate some stressors. However, findings are rather mixed on the protective effects of digital workplace context, as highlighted in a integrative review of the dark side literature by [5]. The construct is variously referred to as technology self-efficacy [65], ICT self-efficacy [64], computer confidence [63], and other similar terms. Here we refer to digital workplace confidence (DWC) meaning workers' ability to apply existing digital skills to all aspects of the digital workplace.

## Research questions and hypotheses

This study aimed to fill the gap in existing research in this domain by looking at the relationships between digital workplace dark side effects, trait mindfulness, digital workplace confidence, and employee burnout and health. It leveraged Job Demands-Resources theory [13] as a theoretical lens through which to investigate the effect of digital workplace dark side effects on employee well-being, as well as any protective benefits that TM and DWC might provide in this context. We make novel contributions to the literature by demonstrating how mindfulness and digital workplace confidence act as personal resources to reduce well-being impacts from the dark side of digital working.

Our research questions were 'To what extent do personal resources of mindfulness and digital workplace confidence protect employee well-being in the digital workplace?' and 'In what ways do these personal resources help to protect well-being in this context?'. Quantitative and qualitative components sought, respectively, to answer these questions.

The study used a sequential explanatory mixed methods approach [66] comprising a cross-sectional survey and semi-structured interviews. Data was collected from workers across different organisations and industries regarding their perceptions of the digital workplace and any associated dark side effects and subsequent well-being impacts, as well as potential mitigating roles for TM and DWC. Based on the prior literature, as explored in this section–and while controlling for age, gender, tenure, work hours per week, education, income and work stress–in the quantitative study we tested the following research hypotheses:

**H1a.** Digital workplace dark side effects are positively predictive of employee burnout and negatively predictive of employee health.

**H1b.** Burnout mediates the relationship between the digital workplace dark side effects and employee health.

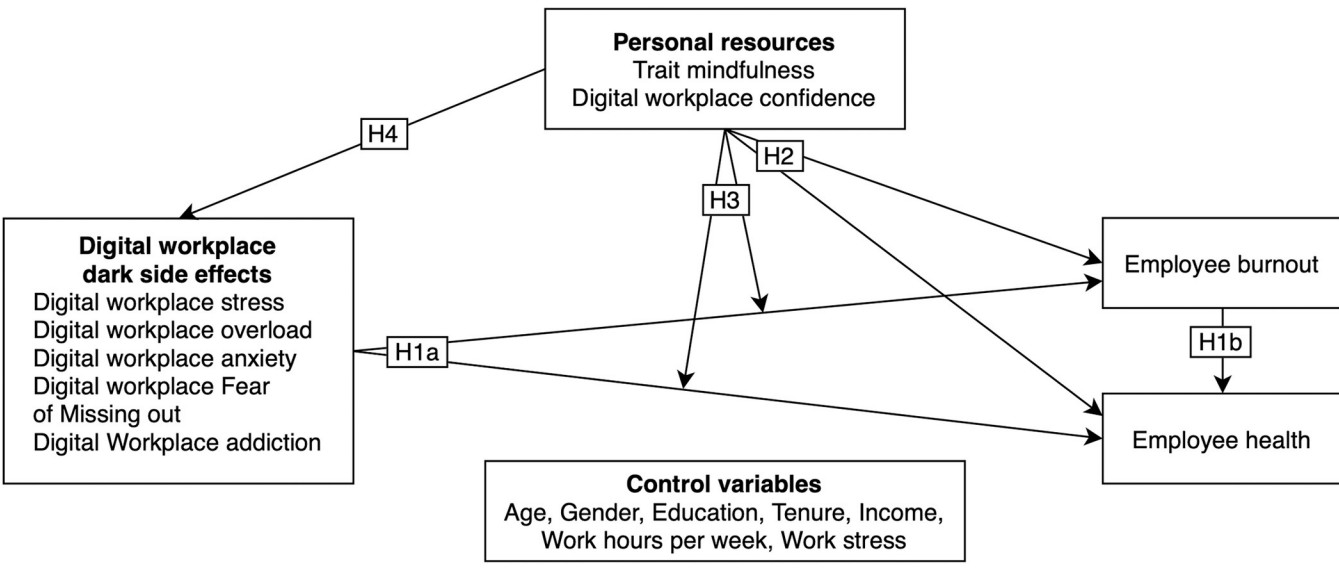

**Fig 1. Research model and hypotheses.**

**H2**. Mindfulness and digital workplace confidence are negatively predictive of employee burnout and positively predictive of health.

**H3**. Mindfulness and digital workplace confidence moderate the relationships between the dark side of digital working effects and employee well-being outcomes (burnout and health), reducing the strength of the relationships.

**H4.** Mindfulness and digital workplace confidence act as negative antecedents of the dark side of digital working effects.

The research model based on these hypotheses is shown in Fig 1.

In the qualitative component, we sought further understanding of how TM and DWC as personal resources may be protective of employee well-being in the digital workplace.

## Materials and methods

### Quantitative methodology

**Design.** The design of the quantitative component of the present study is correlational. Digital workplace demands (manifest through perceived stress, overload, anxiety, Fear of Missing Out and addiction) are included as predictor variables, while burnout and health are outcome variables. Trait mindfulness and digital workplace confidence are included in the model both as moderators and antecedents. The term 'digital workplace' was defined for participants as: 'the collection of digital technologies available in the workplace that enable work to happen, for example: e-mail, intranet, HR systems, productivity tools, mobile devices (etc)'.

**Participants and procedure.** Participants for this study ($N = 142$) were recruited using convenience sampling on the Prolific platform (www.prolific.co) based on certain criteria (UK-based, in full- or part-time work, technology use at work more than once a day). See **Table 2** for demographic characteristics. An a priori power analysis was conducted using G*Power version 3.1.9.6 [67] to determine the minimum sample size required to test the hypotheses. Results indicated that the required sample size to achieve 90% power for detecting a small effect, at a significance criterion of $\alpha = .05$, was $N = 130$ for multiple regression

including seven predictors. Thus, the obtained sample size of $N = 142$ is adequate to test the study hypotheses.

An attention check item was included in which participants were asked to mark a particular response option [68]; two participants failed the attention check and were removed from subsequent analyses. Participation was on a voluntary basis and involved completion of a survey run using Qualtrics (www.qualtrics.com). Participants were provided with a Participant Information Sheet prior to completing a Survey Consent Statement and commencing the survey, as well as a Debrief page immediately following completion. The survey was approved by the University Research Ethics Committee at the University of Nottingham and survey data was collected between 24th and 25th September 2021. Participants completed a questionnaire with items relating to perceived digital workplace demands, trait mindfulness, digital workplace confidence, burnout and health (see **S2 Appendix**). No responses were discarded due to being incomplete. The open-ended question 'Is there anything you would like to add regarding your experience of digital workplace technologies?' was asked at the end of the survey, with responses included within the qualitative data analysis. The survey took an average of 12 minutes to complete and participants were compensated Prolific's recommended amount of £7.50 per hour for surveys. Survey data were analysed in SPSS (Version 26) using linear regression, with variables entered or removed from the model based on a significance level of $\alpha = 0.05$.

**Measures.** Measures of digital workplace dark side effects, trait mindfulness, digital workplace confidence, burnout and health were included. **Table 1** provides an overview of the measures used to investigate participants' perceptions of the constructs included in this study. All variables were coded so that higher values indicated higher levels of that factors, for example, higher values of digital workplace stress indicate greater stress. Descriptive statistics for demographics and work stress are presented in **Table 2** in the Results section.

Demographic controls for age, gender, tenure, work hours per week, education and income were included for their potential to influence the dependent variables. In particular, age and gender have been shown to potentially influence experiences of the dark side effects. For example, feature overload may be more detrimental for older workers while information/ communication overload may be a greater issue for younger workers [74]; and while women may experience more techno-complexity and uncertainty, men may experience more techno-overload and invasion [75]. We also controlled for work stress specifically to rule out an alternate

**Table 1. Measurement scales with Cronbach's Alphas and sample items.**

| Construct | α | Scale | Sample item(s) |
|---|---|---|---|
| Digital workplace stress | .93 | 4 items from the Unified Theory of Acceptance and Use of Technology [69] | 'Working all day with digital workplace technologies is a strain for me.' |
| Digital workplace overload | .82 | Technology overload scale [23] | 'I find that I am overwhelmed by the amount of information I have to process on a daily basis.' |
| Digital workplace anxiety | .76 | 4 items from the Unified Theory of Acceptance and Use of Technology [69] | 'I hesitate to use digital workplace applications for fear of making mistakes I cannot correct.' |
| Digital workplace FoMO | .92 | Fear of Missing Out at work scale [28] | 'When working in the digital workplace. . . I worry that I might miss important work-related updates.' |
| Digital workplace addiction | .78 | 4 items based on [25, 27, 70] | 'I feel I use digital workplace technologies in excess in my life.' |
| Trait mindfulness | .86 | 15 items from the Mindful Attention Awareness Scale [44], selected based on [71] | 'I do jobs or tasks automatically, without being aware of what I'm doing' |
| Digital workplace confidence | .88 | 10 items from the computer self-efficacy scale [61] | 'I could complete the task or job using digital workplace technologies if there was no one around to tell me what to do.' |
| Burnout | .87 | 16 items from the Oldenburg Burnout Inventory [72] | 'There are days when I feel tired before I arrive at work' |
| Health | .81 | 10 items from the Short Form 36 Health Survey Questionnaire [73] | 'I am healthy as anybody I know.' |

**Table 2. Demographic characteristics and work stress.**

| Characteristic | *n* | *%* |
|---|---|---|
| Age | | |
| 18–24 years | 14 | 10.0 |
| 25–34 years | 89 | 63.6 |
| 35–44 years | 22 | 15.7 |
| 45–54 years | 15 | 10.7 |
| Gender | | |
| Male | 22 | 15.7 |
| Female | 118 | 84.3 |
| Education | | |
| Secondary school | 8 | 5.7 |
| Higher education | 35 | 25.0 |
| College/university | 72 | 51.4 |
| Post-graduate degree | 25 | 17.9 |
| Tenure | | |
| Less than 1 year | 19 | 13.6 |
| 1–4 years | 64 | 45.7 |
| 5–9 years | 33 | 23.6 |
| 10–19 years | 19 | 13.6 |
| 20–29 years | 4 | 2.9 |
| Work hours per week | | |
| 16 hours or less | 6 | 4.3 |
| 17–24 hours | 13 | 9.3 |
| 25–40 hours | 96 | 68.6 |
| More than 40 hours | 25 | 17.9 |
| Income | | |
| Less than £10,000 | 10 | 7.1 |
| £10,001–20,000 | 27 | 19.3 |
| £20,001–30,000 | 50 | 35.7 |
| £30,001–40,000 | 32 | 22.9 |
| £40,001–50,000 | 11 | 7.9 |
| More than £50,000 | 10 | 7.1 |
| Work stress | | |
| Not at all stressful | 7 | 5.0 |
| Mildly stressful | 34 | 24.3 |
| Moderately stressful | 69 | 49.3 |
| Very stressful | 25 | 17.9 |
| Extremely stressful | 5 | 3.6 |

*N* = 140.

explanation for digital workplace stress and burnout: [76] identified a direct association between technostress and role stress.

## Qualitative methodology

**Design.** In addition to the cross-sectional survey, semi-structured interviews were conducted to understand how personal resources of mindfulness and digital workplace confidence may help to reduce the dark side of digital working.

**Participants and procedure.** Interview participants ($N$ = 14) were working individuals in the UK who use computers at work. They were recruited via convenience sampling on Prolific. Participants were asked if they experienced stress, overload, anxiety or excessive/ compulsive use in relation to digital workplace technologies, and whether any of these experiences have led them to feel exhausted in their work. They were asked about their perceptions of whether TM and DWC help to counteract any adverse effects experienced in the digital workplace and if so in what way. Interview questions were created based on theoretical concepts and measurement instruments used in the survey (see previous section). The semi-structured nature of the interviews afforded participants the opportunity to freely explore and discuss the topics, allowing for an in-depth understanding of their perspectives.

Participants were provided with a Participant Information Sheet and signed an Interview Consent Statement before participating in a recorded interview lasting no more than 45 minutes via Microsoft Teams or telephone. Participants were compensated Prolific's recommended amount of £10.00 per hour for interviews. Following the interview, participants were provided with information to support mental health at work.

The interview procedure was approved by the University Research Ethics Committee at the University of Nottingham and interviews were conducted between 29th June and 21st July 2022. Interviewees were aged between 27 and 60 years with tenures in their current organisation ranging from under 1 year to 20 years. Eight women and 6 men were interviewed. Participants fulfilled a wide range of roles including: teacher, operations manager, learning disabilities worker, nuclear health physicist, logistics support administrator, store manager, and solicitor. The sample, though relatively small [77], was considered to have relatively high information power [78]; in particular, high sample specificity, strong theoretical support, and strong dialogue with participants) and fulfilled the needs of the explanatory sequential mixed methods approach in which the qualitative data help to elucidate the quantitative results [66].

Anonymisation of data was carried out during transcription; transcripts were analysed in NVIVO. Initial codes were derived from the personal resources constructs in the quantitative study; additional codes were identified inductively during review and analysis [66]. Codes ($N$ = 15) were then grouped into themes ($N$ = 3).

Prior to data collection, this study was pre-registered on Open Science Framework (https://osf.io/f9ymb) including details of the study design, sampling, variables, and analysis plan.

## Results

### Quantitative findings

**Descriptive statistics.** Descriptive statistics are reported in Table 2. The majority of respondents were female (n = 118: 84.3%), aged 25–34 years (n = 89; 63.6%) and college/university educated (n = 97; 69.3%). Most had a tenure of 1–9 years (n = 97; 69.4%) in their present company and worked 25–40 hours per week (n = 96; 68.6%).

**Common method bias.** Common method bias (CMB) can be an issue for self-report data, potentially distorting the relationships between the variables by inflating the correlations [79]. The potential effect of CMB was checked using Harman's single factor test. The result of the test showed that the total variance extracted by one factor is 41.99% which is below the recommended threshold of 50% [80] meaning that CMB is not a problem in the current data set.

**The effect of the dark side effects on burnout and health.** The relationships the digital workplace dark side effects have with employee outcomes of burnout and health were explored via linear regression. Two forced entry linear regressions to predict burnout and health respectively were conducted with demographics and work stress entered at step 1, and the dark side

effects entered at step 2. Pearson's correlations are provided alongside for comparison. All parametric assumptions were met.

When burnout was included as the employee outcome, Step 1 variables of demographics and work stress accounted for 26% of the variance ($R^2$ = .26, $F$ (7, 131) = 6.7, $p < .01$); work stress made a statistically significant unique contribution. Digital workplace dark side effects were incorporated into the model at Step 2 and as a result a significant increase in variance explained of 17% ($\Delta R^2$ = .17, $F$(5, 126) = 7.58, $p < .01$) was observed, although only digital workplace stress was significant, positively predicting burnout (see **Table 3**). Hypothesis 1a was therefore partially supported. Although other dark side effects did not reach significance in the regression, they all had significant positive correlations with burnout, indicating that the variance they explain individually may overlap.

When health was the outcome, demographics and work stress accounted for 15% of variance ($R^2$ = .15, $F$(7, 131) = 3.37, $p < .01$); work stress was again a significant predictor. Dark side effects also significantly predicted 13% further variance ($\Delta R^2$ = .13, $F$(5, 126) = 4.56, $p < .01$). Digital workplace stress and FoMO negatively predicted health (see **Table 4**), partially supporting Hypothesis 1a. Again, all dark side effects, except addiction, were negatively and significantly correlated with health.

Mediation analyses (and subsequent moderation analyses), were performed using the PROCESS v4.0 module in SPSS [81] using ordinary least squares path analysis. Models 4 and 1 were used to test the mediation and moderation hypotheses, respectively. Bootstrap confidence intervals with 5,000 samples were used to estimate indirect effects [82]. Mediation was explored to see if the effects of the digital workplace dark side effects were transmitted onto

**Table 3. Predicting burnout from the dark side effects.**

|  |  | r | B (SE) | t |
|---|---|---|---|---|
| Step 1 | Age | -.17* | -.06 (.10) | -.70 |
|  | Gender | .07 | -.04 (.20) | -.46 |
|  | Education | .07 | .05 (.10) | .60 |
|  | Tenure | .00 | .02 (.08) | .26 |
|  | Income | -.20* | -.20 (.07) | -2.03* |
|  | Work hours per week | -.07 | -.10 (.13) | -1.04 |
|  | Work stress | .44** | .47 (.08) | 5.76** |
| Step 2 | Age | - | -.07 (.09) | -.81 |
|  | Gender | - | -.06 (.18) | -.88 |
|  | Education | - | .13 (.09) | 1.75 |
|  | Tenure | - | .00 (.07) | .015 |
|  | Income | - | -.09 (.06) | -1.04 |
|  | Work hours per week | - | -.14 (.11) | -1.72 |
|  | Work stress | - | .28 (.08) | 3.45** |
|  | Digital workplace stress | .49** | .30 (.07) | 3.65** |
|  | Digital workplace overload | .42** | .14 (.09) | 1.61 |
|  | Digital workplace anxiety | .38** | .11 (.06) | 1.36 |
|  | Digital workplace FoMO | .40** | .07 (.06) | .74 |
|  | Digital workplace addiction | .15* | -.03 (.06) | -.36 |

\* $p < .05$

\** $p < .01$

$r$ = Pearson's correlation coefficient, $B$ = standardized beta coefficients, $SE$ = standard error, $t$ = test of statistical significance. $N$ = 142, with pairwise deletion for missing data.

**Table 4. Predicting health from the dark side effects.**

| | | Step 1 | | Step 2 | |
|---|---|---|---|---|---|
| | *r* | *B (SE)* | *t* | *B (SE)* | *t* |
| Age | .27** | .11 (.10) | 1.18 | .15 (.10) | 1.64 |
| Gender | -.05 | -.00 (.19) | -.04 | .04 (.18) | .52 |
| Education | .06 | .07 (.09) | .81 | -.01 (.09) | -.13 |
| Tenure | .31** | .25.(08) | 2.83** | .24 (.07) | 2.80** |
| Income | .19* | .15 (.07) | 1.47 | .04 (.07) | .35 |
| Work hours per week | -.00 | -.07 (.12) | -.73 | -.02 (.12) | -.19 |
| Work stress | -.10 | -.11 (.08) | -1.30 | .05 (.09) | .60 |
| Digital workplace stress | -.32** | - | - | -.24 (.07) | -2.61* |
| Digital workplace overload | -.20** | - | - | -.02 (.09) | -.23 |
| Digital workplace anxiety | -.21** | - | - | -.01 (.07) | -.14 |
| Digital workplace FoMO | -.37** | - | - | -.25 (.06) | -2.45* |
| Digital workplace addiction | -.07 | - | - | .08 (.06) | 1.00 |

\* *p* < .05

\*\* *p* < .01

*r* = Pearson's correlation coefficient, *B* = standardized beta coefficients, *SE* = standard error, *t* = test of statistical significance. *N* = 142, with pairwise deletion for missing data.

health via burnout while controlling for demographics and work stress All dark side effects, except addiction, had a significant indirect effect on health via burnout (see **Table 5** and **S3 Appendix**). Using the standardised indirect effect as an indication of effect size [83], the effect size for all mediations can be considered small [84].

Hypothesis 1b which states that digital workplace dark side effects are predictive of employee health via burnout is therefore partially supported.

**The effect of trait mindfulness and digital workplace confidence on burnout and health.** The relationships that TM and DWC have with employee outcomes of burnout and health were explored via linear regression. Two forced entry linear regressions to predict burnout and health outcomes respectively were conducted with demographics and work stress entered at step 1, and TM and DWC entered at step 2. Pearson's correlations were also examined within the same variables and are provided alongside the regression coefficients in each data table so that direct and partial relationships can be compared. All parametric assumptions were met.

Both TM (*r* = -.53) and DWC (*r* = -.35) were negatively and significantly correlated with burnout. Demographics and work stress accounted for 26% of the variance in burnout ($R^2$ = .26, $F(7, 131)$ = 6.7, *p* < .01), with work stress making a significant unique contribution. TM

**Table 5. Mediation results for the dark side effects on health via burnout.**

| Mediator | Indirect effect | Lower | Upper | Standardised indirect effect ($K^2$) | Upper | Lower |
|---|---|---|---|---|---|---|
| Digital workplace stress | -.10 | -.18 | -.03 | -.13 | -.23 | -.04 |
| Digital workplace overload | -.11 | -.19 | -.04 | -.12 | -.21 | .04 |
| Digital workplace anxiety | -.07 | -.14 | -.02 | -.10 | -.19 | -.03 |
| Digital workplace FoMO | -.06 | -.12 | -.02 | -.09 | -.18 | -.03 |
| Digital workplace addiction | -.02 | -.07 | .02 | -.03 | -.11 | .04 |

Bootstrap analyses (5000 bootstrap samples); 95% confidence intervals.

**Table 6. Predicting burnout from trait mindfulness and digital workplace confidence.**

| | | Step 1 | | Step 2 | |
|---|---|---|---|---|---|
| | r | B (SE) | t | B (SE) | t |
| Age | -.17* | -.06 (.10) | -.70 | -.00 (.09) | -.05 |
| Gender | .07 | -.04 (.20) | -.46 | -.07 (.17) | -.94 |
| Education | .07 | .05 (.09) | .60 | .12 (.08) | 1.6 |
| Tenure | .00 | .02 (.08) | .26 | .13 (.07) | 1.7 |
| Income | -.20** | -.20 (.07) | -2.03* | -.19 (.06) | -2.17* |
| Work hours per week | -.07 | -.10 (.13) | -1.04 | -.06 (.11) | -.68 |
| Work stress | .44** | .47 (.08) | 5.76** | .28 (.08) | 3.73** |
| Trait mindfulness | -.53** | - | - | -.44 (.08) | -5.83** |
| DWC | -.35** | - | - | -.16 (.05) | -2.22* |

\* $p < .05$

\*\* $p < .01$

$r$ = Pearson's correlation coefficient, $B$ = standardized beta coefficients, $SE$ = standard error, $t$ = test of statistical significance. $N$ = 140, with pairwise deletion for missing data.

and DWC were incorporated into the model at Step 2 and as a result a significant increase in variance explained ($\Delta R^2$) of 19% was observed ($\Delta R^2$ = .19, $F(2, 129)$ = 23.0, $p < .01$). Both TM and DWC negatively predicted burnout (see **Table 6**), supporting Hypothesis 2.

The relationship between TM and burnout ($b$ = -.44, $p < .001$, 95% CI [-.63, -.31]) had a higher beta value than the relationship between DWC and burnout ($b$ = -.16, $p < .05$, 95% CI [-.19,-.01]), and the difference in the strength of relationships observed was significant: the confidence intervals do not overlap by more than 50% [85]. Workers with higher digital workplace confidence and, even more so, those who were more mindful experienced less burnout at work.

Both TM ($r$ = .42) and DWC ($r$ = .28) were positively and significantly associated with health. When health was included in the regression model as the employee outcome, Step 1 variables of demographics and work stress accounted for 15% of the variance ($R^2$ = .15, $F(7, 131)$ = 3.37, $p < .01$). TM and DWC were incorporated into the model at Step 2, increasing variance explained significantly by 13% ($\Delta R^2$ = .13, $F(2, 129)$ = 11.73, $p < .01$), with both TM and DWC positively predicting health (see **Table 7**), lending further support to Hypothesis 2.

**Table 7. Predicting health from trait mindfulness and digital workplace confidence.**

| | | Step 1 | | Step 2 | |
|---|---|---|---|---|---|
| | r | B (SE) | t | B (SE) | t |
| Age | .27** | .12 (.10) | 1.18 | .09 (.10) | .98 |
| Gender | -.05 | -.00 (.19) | -.04 | .01 (.18) | .16 |
| Education | .06 | .07 (.09) | .81 | .02 (.09) | .28 |
| Tenure | .31** | .25 (.08) | 2.83* | .18 (.07) | 2.06* |
| Income | .19* | .15 (.07) | 1.47 | .12 (.06) | 1.25 |
| Work hours per week | -.00 | -.07 (.12) | -.73 | -.09 (.12) | -1.01 |
| Work stress | -.10 | -.11 (.08) | -1.30 | .05 (.08) | .52 |
| Trait mindfulness | .42** | - | - | .30 (.09) | 3.52** |
| DWC | .28** | - | - | .21 (.05) | 2.55* |

\* $p < .05$

\*\* $p < .01$

$r$ = Pearson's correlation coefficient, $B$ = standardized beta coefficients, $SE$ = standard error, $t$ = test of statistical significance. $N$ = 140, with pairwise deletion for missing data.

Although the relationship between TM and health ($b = .30$, $p < .001$, 95% CI [.13, .47]) again had a higher beta value than the relationship between DWC and health ($b = .21$, $p < .05$, 95% CI [.03, .22]), this difference was not significant [85].

**Trait mindfulness and digital workplace confidence as moderators between dark side effects and employee well-being outcomes.**   TM and DWC were hypothesised to moderate the effect of the dark side effects on burnout and health while controlling for demographics and work stress. Following the results of the regressions, moderation effects were only tested in the cases of digital workplace stress on burnout/ health, and digital workplace FoMO on health. Items were mean centred before the moderation analysis to reduce collinearity [86].

The interaction terms between digital workplace stress and, firstly, TM and, secondly, DWC were added to the regression model, which accounted for a significant proportion of the variance in burnout ($R^2 = .48$, $F(5, 134) = 20.04$, $p < .01$). However, neither interaction was significant. When the relationship between digital workplace stress and health was explored in the same manner, the model again accounted for a significant proportion of the variance in health ($R^2 = .33$, $F(12, 126) = 5.14$, $p < .001$) but again neither interaction was significant.

The interaction terms between digital workplace FoMO and, firstly, TM and, secondly, DWC were added to the regression model, which accounted for a significant proportion of the variance in health ($R^2 = .31$, $F(12, 126) = .56$, $p < .01$). However, neither interaction was significant.

Hypotheses 3, stating that TM and DWC will moderate the relationship between the dark side of digital working effects and employee well-being outcomes, was therefore not supported.

**Trait mindfulness and digital workplace confidence as antecedents of the dark side effects.**   Five forced entry linear regressions were conducted with demographics and work stress entered at step 1, and TM and DWC entered at step 2. Demographics and work stress accounted for between 11% and 23% of variance in each of the dark side effects (see **Table 8**). Work stress made statistically significant unique contributions to all dark side effects except addiction.

TM and DWC were incorporated into the model at Step 2 and significantly negatively predicted digital workplace stress ($\Delta R^2 = .09$, $F(2, 129) = 7.41$, $p < .01$), overload ($\Delta R^2 = .13$, $F(2, 129) = 10.90$, $p < .01$), anxiety ($\Delta R^2 = .18$, $F(2, 129) = 16.83$, $p < .01$) and FoMO ($\Delta R^2 = .15$, $F(2, 129) = 15.28$, $p < .01$): between 9% and 18% of variance in constructs was explained. The model relating to digital workplace addiction ($\Delta R^2 = .04$, $F(2, 129) = 2.96$, ns) was not significant.

TM was negatively and significantly correlated with all dark side of digital working effects, with small to medium effect sizes. TM made statistically significant unique contributions to all the dark side effects supporting hypothesis 4: more mindful individuals were less likely to experience digital workplace stress ($b = -.31$, $p < .01$), overload ($b = -.41$, $p < .01$), anxiety ($b = -.33$, $p < .01$), FoMO ($b = -.41$, $p < .01$) and addiction ($b = -.21$, $p < .05$).

DWC was negatively and significantly correlated with digital workplace stress, anxiety and FoMO. In the regression, it significantly predicted just one of the five dark side effects: digital workplace anxiety, partially supporting Hypothesis 4. Workers who were more digitally confident were less likely to experience digital workplace anxiety ($b = -.27$, $p < .01$).

Betas for the dark side effects were higher for TM than for DWC. This difference was significant in the case of all dark side effects except for anxiety: the confidence intervals do not overlap by more than 50% [85]. In other words, TM was more effective than DWC for reducing digital workplace stress, overload, FoMO and addiction (but not anxiety).

**Table 8. Predicting the digital workplace dark side effects from trait mindfulness and digital workplace confidence.**

| | | Digital workplace stress | | | Digital workplace overload | | | Digital workplace anxiety | | | Digital workplace FoMO | | | Digital workplace addiction | | |
|---|---|---|---|---|---|---|---|---|---|---|---|---|---|---|---|---|---|
| | | *r* | *B (SE)* | *t* | *r* | *B (SE)* | *t* | *r* | *B (SE)* | *t* | *r* | *B (SE)* | *t* | *r* | *B (SE)* | *t* |
| Step 1 | Age | -.07 | .01 (.13) | .06 | -.08 | -.03 (.11) | -.32 | -.12 | -.05 (.14) | -.47 | -.12 | .10 (.15) | 1.07 | -.20* | -.15 (.14) | -1.56 |
| | Gen | .02 | .02 (.26) | .18 | .02 | -.01 (.22) | -.06 | .15* | .11 (.28) | 1.26 | .19* | .17 (.29) | 2.05* | .06 | -.00 (.28) | -.03 |
| | Edu | -.11 | -.16 (.12) | -1.88 | -.05 | -.10 (.10) | -1.09 | -.08 | -.08 (.13) | -.91 | -.07 | -.13 (.14) | -1.57 | .16* | .09 (.13) | .98 |
| | Ten | .07 | .05 (.10) | .57 | .03 | .02 (.08) | .26 | .05 | .08.(11) | .88 | -.11 | -.14 (.11) | -1.61 | -.15* | -.11 (.11) | -1.16 |
| | Inc | -.09 | -.17 (.09) | -1.65 | -.06 | -.06 (.08) | -.53 | -.25** | -.22 (.10) | -2.09* | -.24** | -.27 (.10) | -2.71** | .08 | .08 (.10) | .77 |
| | Hrs | .13 | .15 (.17) | 1.57 | .03 | -.01 (.14) | -.05 | -.10 | -.02 (.18) | -.21 | -.01 | .08 (.18) | .83 | .12 | .02 (.18) | .22 |
| | WS | .34** | .35 (.11) | 4.02** | .30** | .32 (.09) | 3.64** | .20* | .22 (.12) | 2.44* | .32** | .36 (.12) | 4.35** | .22** | .17 (.12) | 1.89 |
| Step 2 | Age | - | .05 (.13) | .57 | - | .06 (.11) | .58 | - | -.03 (.13) | -.31 | - | .16 (.14) | 1.89 | - | -.09 (.15) | -.89 |
| | Gen | - | -.01 (.25) | -.08 | - | -.04 (.20) | -.46 | - | .09 (.25) | 1.18 | - | .14 (.26) | 1.87 | - | -.02 (.27) | -.27 |
| | Edu | - | -.12 (.12) | -1.41 | - | -.04 (.10) | -.47 | - | -.03 (.12) | -.35 | - | -.07 (.13) | -.93 | - | .12 (.13) | 1.29 |
| | Ten | - | .12 (.10) | 1.38 | - | .11 (.08) | 1.24 | - | .17 (.10) | 2.03* | - | -.05 (.10) | -.58 | - | -.07 (.11) | -.76 |
| | Inc | - | -.17 (.09) | -1.70 | - | -.09 (.07) | -.84 | - | -.18 (.09) | -1.84 | - | -.27 (.09) | -2.96** | - | .04 (.10) | .40 |
| | Hrs | - | .19 (.16) | 1.97 | - | .05 (.13) | .49 | - | -.00 (.16) | -.01 | - | .12 (.17) | 1.39 | - | .06 (.18) | .56 |
| | WS | - | .23 (.11) | 2.57* | - | .20 (.09) | 2.27* | - | .03 (.12) | .33 | - | .21 (.12) | 2.54* | - | .14 (.13) | 1.47 |
| | TM | -.35** | -.31 (.12) | -3.49** | -.42** | -.41 (.09) | -4.62** | -.40** | -.33 (12) | -3.93** | -.49** | -.41 (.12) | -5.05** | -.26** | -.21 (.13) | -2.25* |
| | DWC | -.23** | -.07 (.07) | -.89 | -.14 | .03 (.06) | .30 | -.37** | -.27 (.07) | -3.39** | -.28** | -.09 (.07) | -1.20 | .04 | .12 (.07) | 1.36 |
| | | $R^2$ = .17 for Step 1; $\Delta R^2$ = .09 for Step 2. | | | $R^2$ = .11 for Step 1; $\Delta R^2$ = .13 for Step 2. | | | $R^2$ = .13 for Step 1; $\Delta R^2$ = .18 for Step 2. | | | $R^2$ = .23 for Step 1; $\Delta R^2$ = .15 for Step 2. | | | $R^2$ = .11 for Step 1; $\Delta R^2$ = .04 ns for Step 2. | | |

\* $p < .05$

\*\* $p < .01$

*r* = Pearson's correlation coefficient, *B* = standardized beta coefficients, *SE* = standard error, *t* = test of statistical significance. *N* = 140, with pairwise deletion for missing data. Gen = Gender; Edu = Education; Ten = Tenure; Inc = Income; Hrs = Work hours; WS = Work stress; TM = Trait mindfulness; DWC = digital workplace confidence.

## Qualitative findings

Codes relating to the personal resources of mindfulness and digital workplace confidence were organised into three overarching themes (see **Table 9**).

**Digital workplace confidence supports sense of agency.** Based on comments about their DWC during the interviews, six participants could be considered to have a high level of confidence using the digital workplace, while for the remaining eight it was medium (n = 5) or low (n = 3). Higher digital workplace confidence tended to equate to a greater sense of agency when using technology for work:

**Table 9. Summary of themes.**

| Theme | Description |
|---|---|
| Digital workplace confidence supports sense of agency | Higher DWC enabled control and effectiveness in the digital workplace while lower DWC could lead to disempowerment, though training/support could help counteract this. |
| Reflection and learning are key to changing digital habits | Reflecting on the impact of technology on personal well-being and having a learning mindset in relation to it enable individuals to change digital habits for the better and set boundaries in the digital workplace. |
| Digital mindfulness can help protect well-being | A mindful orientation towards technology was seen as potentially helpful in avoiding negative well-being impacts; some engage in mindful practices such as breathing, taking a break, and checking in with their mental and physical state. |

*"I'm always very much in control and you know I'm making things happening with technology, I don't feel that technology is driving me."* [P14]

By contrast, those with lower DWC could feel forced to use technology against their will or driven by the technology to work in a certain way:

*"You have to use [technology] at a certain time of day, so all of a sudden in a way you're becoming driven by the technology rather than you driving it."* [P6]

*"Sometimes you are forced to just using [technology] and it has to be done and that's the stressful part of it."* [P2]

For those least confident with the technology, they could at times feel disempowered by it.

*"Sometimes you just feel a bit useless [. . .] why is this so challenging for me? [. . .] you feel completely disempowered."* [P11]

One manager suggested that such experiences might even lead people to quit their jobs:

*"Somebody that used to work on my team that really struggled with [technology] and I know she used to get stressed out with it because [. . .] she couldn't get a grip of it. [. . .] people have left in the past as well so I'm sure that it has an impact on their self-esteem."* [P4]

There were tactics among those with lower DWC to try to take back the sense of agency that was potentially lost when working digitally. Sometimes this took the form of accessing training or support (although having to constantly ask for help could lead back to feeling disempowered).

*"I've been very lucky to be supported by a good workplace and a good support network, whether it be IT or work colleagues that know what they're doing, which has reduced any sort of stress or anxiety that I would have perhaps felt otherwise."* [P13]

Another tactic was to seek out a quiet space (e.g., alone in a meeting room or at home) to deal with technology and at a time that is less pressured (e.g., after work hours).

*"Because I had the time to just explore [the technology] a little bit or because I received some training I feel more confident."* [P2]

It was noticeable that digital workplace anxiety tended to be higher for this group (e.g., about using new technologies or about losing data or documents) and these tactics appeared to help them manage this.

**Reflection and learning are key to changing digital habits.**   Among participants with higher DWC, in particular, reflecting on the impact that technology can have on well-being was an important tool for learning about and changing their digital habits.

*"I think it's important that in the first instance we are aware of the technology we're using, how it's impacting us, and then maybe set some strategies to make sure that that impact is more positive."* [P9]

A few of the participants shared that the reflection involved in participating in the interview was leading them to review their digital practices:

*"I think with this talk I just take with me so many things that I know I have done good from what I've learned, but I think I would take other things that I haven't been doing and I would definitely apply in the future and I think I hadn't realized the stress that it could cause me."* [P3]

Reflection and learning–either from personal experiences of using technology in a certain way, or through dedicating time to learning new applications or features–could form the basis for changing digital habits for the better.

*"I think it's just changing habits. Basically, just being conscious about what affects me and then seeing how I can change it so it's good for me and it's good for my colleagues."* [P3]

For one participant this involved noticing how messaging overload had contributed to burnout and changing the way he dealt with emails as a result:

*"I've probably been [batching emails] for about the last 3 years. Before that I did have notifications turned on and would try to respond immediately to people trying to get hold of me by email."* [P14]

Such changes tended to involve setting boundaries in the digital workplace by implementing specific rules for how and when to engage in the digital workplace (e.g., turning off notifications, shutting down devices, protecting personal contact information, turning off the camera).

*"On my phone I don't have notifications come through from apps [. . .] it prevents that time wastage element of apps."* [P10]

*"Shutting down the computer [. . .] and just closing the door to the room and not going back in if I can help it."* [P1]

It was noticeable that those with higher DWC were more likely to leverage the affordances of the technology to help set boundaries (e.g., turning off notifications), while those with lower DWC were more likely to retreat from the technology in order to enforce a boundary (e.g., putting devices in another room). This connects to the sense of agency felt by those with higher DWC that we saw in the first theme.

**Digital mindfulness can help protect well-being.** Most participants felt that a mindful orientation toward technology could help to regulate use in a way that would help them to avoid negative well-being impacts. Linking to the previous theme, such an orientation could help them become more aware of their technology usage and put boundaries in place. It could also help them to feel more 'grounded' and in the moment; not get distracted or 'carried away' by technology; be able to focus on work more fully; and get off autopilot mode;

*"I think it shouldn't be that way [multitasking], especially when it comes to technology, as it's something that it builds up [. . .] so I think it would be very important to be present while you're working with [technology]."* [P3]

*"Getting yourself to think [. . .] Am I mentally OK? [. . .] How am I feeling physically? Because like I say there are days when I sit there all day and then at the end of the day it can be gosh*

*my back really hurts. [. . .] And [being mindful] would enable someone to pick that up earlier on in the day."* [P4]

Some felt that mindfulness could be helpful in a technologically enabled work environment but that it could be hard to engage with due to the lack of a quiet space, time pressures and managing perceptions of cyber slacking.

*"If it was during the working day, I wouldn't log out of Jabber or anything to, you know, be offline for a while [. . .] because I don't want people to think: Where's she gone?"* [P1]

While not necessarily engaging in formal mindfulness practice, some of the participants reported acting mindfully to help protect well-being in the digital workplace. This could be as simple as taking a breath or having a break from technology, as well as checking in with their own mental and physical state of being.

*"It does help if you can switch off from [technology]. And I think you should get up from your computer and go off and have a wander and make a cup of tea. Go and have a chat with a colleague or whatever. [. . .] Do some human things."*

*"Sometimes if it's overwhelming I take a few deep breaths for a few seconds and pause before I carry on."* [P5]

One participant said that her Christian faith gave her a mindful orientation towards technology:

*"I'm often quite conscious of [. . .] wanting to make sure that I'm trying to live as healthy a Christian life as possible. And so things like the way I use technology and the way that I try to think about technology, I would describe it as like no one needs to be like an idol in my life."* [P10]

Operating on autopilot, though necessary to a certain degree, is contrasted with the present moment awareness that characterises mindfulness. Autopilot was particularly associated with doing routine and repetitive tasks in the digital workplace rather than collaborating with others or tackling complex or novel tasks.

*"You're obviously literally on autopilot doing this and without realising you completed it."* [P5]

Automaticity was seen as a positive where it implied that the individual was comfortable with the technology and could complete the task without too much effort; but it was also seen as negative where it led to lack of awareness of the quality of work done, working for too long or lack of physical movement while working.

*"Sometimes I can have been on my computer or I will see sometimes that time has gone for half an hour, and I can't always tell you exactly what I've been doing. I know I would have been working and doing things."* [P10]

## Discussion

In this study, Job Demands-Resources model [15] provided the foundation for an exploration of personal resources of trait mindfulness and digital workplace confidence as mitigating

harms to employee well-being due to the dark side of digital working. Our results show that while digital workplace stress and FoMO, in particular, were found to be associated with higher burnout and poorer health, both TM and DWC protected employee well-being by acting as antecedents: TM for all dark side effects, and DWC for digital workplace anxiety and FoMO. Data demonstrating impacts on health due to an array of dark side effects and resulting burnout are novel; the protective effects of two specific personal resources is also a new, important addition to the literature, with TM showing a greater effect than DWC in this respect. Interview data indicated that a mindful orientation to the digital workplace could help minimise negative well-being impacts; while being more digitally confident supported a sense of agency and changing digital habits for the better. We extend understanding of impacts to employee health in the digital workplace; clarify the role of DWC in relation to anxiety-related aspects of digital work; and reveal a potentially important role for mindfulness in helping to protect employee well-being in the digital workplace.

Our findings reveal concerning implications for employees increasingly reliant on the digital workplace to facilitate their work: all dark side effects (except addiction) were significantly correlated with both burnout (moderate effects) and health (small effects). In the regressions, digital workplace stress and FoMO made significant unique contributions to burnout and health, respectively. In addition, FoMO negatively impacted health via burnout. The effect of digital workplace stress on burnout aligns with previous research in this domain, while the findings relating to health (so far but scantly covered in the dark side literature) provide new insights [5].

Social media-related FoMO has been found to be associated with physical and mental health impacts [87]. This study extends such findings into a workplace context, suggesting that employees' fears about missing out on informational and relational opportunities in the digital workplace may have negative health implications. Job demands relating to the dark side of digital working–stress and FoMO in particular—should be considered as serious potential hazards to employees, alongside other psychosocial risks in the workplace.

Mindfulness is a personal resource that can help reduce burnout and have positive health benefits at work [88, 89]. Our findings support these associations in the existing literature and extend them into the digital workplace context. TM was negatively correlated with all dark side of digital working effects (small to moderate effects) as well as burnout and health (moderate effects). Furthermore, in the regressions TM made significant unique contributions to burnout, health and all dark side effects except addiction. The qualitative data are in alignment, suggesting that a mindful orientation towards technology can help to reduce negative well-being impacts; as one interviewee [P5] described: "*Sometimes if it's overwhelming I take a few deep breaths for a few seconds and pause before I carry on.*".

Overall, more mindful employees appear to experience less adverse effects in the digital workplace and have better well-being outcomes. Further analyses showed that TM worked as an antecedent of the dark side effects rather than a moderator between them and well-being. This suggests that mindfulness may help protect well-being by altering employees' perceptions of digital workplace stressors, rather than changing the relationship between these stressors and burnout/ health. Indeed, by changing the negative stress appraisal process, mindfulness has been found to help reduce burnout due to workplace stress generally [59] and technology-related stress specifically [90].

While computer confidence may be seen as requisite for reducing stress and strain in the digital workplace [10], findings on its effectiveness in the dark side literature are so far mixed [5]. Results from the present study clarify its role, also extending understanding to burnout and health. DWC was significantly and negatively correlated with digital workplace stress,

anxiety and FoMO, and in our regression model it made statistically significant unique contributions to the latter two of these dark side effects.

More digitally confident workers were less likely to experience digital workplace anxiety, a finding that aligns with the view that computer self-efficacy is a correlate of computer anxiety [27]. They were also less likely to experience FoMO, a novel finding that may be plausibly explained by greater computer confidence helping employees engage effectively with information and relationships in the digital workplace thereby reducing a sense of missing out. Indeed, employees with lower computer competence and confidence [91, 92] are less likely to adopt and use information and communication technologies in the workplace, which could in turn lead to greater fears about missing out on information and connections. Qualitative findings in the present study elucidate the differential effects of high and low DWC: while the former was linked to greater agency through positive engagement with the technology, the latter could mean greater anxiety, a sense of disempowerment and withdrawal from it. An apt example of this contrast was evident in the excerpts from [P14] who said "*I don't feel that technology is driving me*", and [P6] who said: *"[. . .] you're becoming driven by the technology rather than you driving it.".*

Although self-efficacy relating to technology has been found to reduce stress in technology environments [63, 93], this was not entirely borne out by the present findings. Digital workplace confidence was positively correlated with digital workplace stress (small effect) but did not reach significance in the regression, perhaps obscured by the stronger effect of TM. A potential indication from the qualitative data is that the stressful effect of DWC is indirect, transmitted via anxiety: participants with lower DWC tended to experience greater anxiety and find the digital workplace more stressful.

Prior research has indicated the potentially protective effects of the personal resources of TM [88] and DWC [16]; the present study aligns with these findings while elucidating their specific role in a digital work environment. When looking at the comparative effects of TM and DWC on the dark side effects, mindfulness had consistently larger effects (except on anxiety). A context-specific trait such as DWC might be expected to have higher explanatory power than a broad trait such as mindfulness [94]. It is therefore noteworthy that being more mindful appears to be more important than being digitally confident when it comes to protecting employee well-being in the digital workplace. The qualitative data may go some way to elucidate this, as a more mindful orientation toward the technology appeared to be protective of well-being both directly by reducing negative impacts and indirectly by supporting the reflection and learning needed to change digital habits for the better and set boundaries in the digital workplace. This is, to the authors' knowledge, the first study to compare the effects of TM and DWC on dark side effects. The evidence for mindfulness as a personal resource to help reduce employee burnout in context of a range of job demands is growing [95]; the present study adds to this in the specific domain of digital work.

## Limitations and directions for future research

While conducted with theoretical and methodological rigour, findings from the present study should be considered in the light of certain limitations. Survey data was collected at a single point in time, thereby limiting causal interpretation. The survey sample was fairly homogenous (84.3% female) and some small differences have been found with respect to gender and the dark side effects [75]. A large proportion of survey participants (63.6%) were aged 25–34 years; younger workers may experience lower digital workplace dark side effects [63] and so larger effects may have been observed with older workers. The proportion of variance explained in the various regressions (between 14.5% and 45.7%) suggest that other factors not

considered were involved; indeed future studies may benefit from a more homogenous sample in terms of job characteristics and frequency of technology use, perhaps drawn from a single organisation or team. Ideally such studies should be longitudinal in design in order to throw light on causal relationships. In the case of the qualitative data, the sample size limits generalisability; future research should aim to engage larger and more diverse samples in order to further explore these constructs.

Weak quantitative findings in relation to digital workplace addiction, contrasted with issues expressed by interviewees in the area of maladaptive use of the digital workplace, strongly suggest an opportunity to explore this construct using a different definition and measurement instrument. Potential protective effects of digital competence as well as confidence for wellbeing in the digital workplace warrant further investigation. Finally, the present study underlines the Job Demands-Resources model as a fruitful avenue of enquiry in this domain and TM and DWC as personal resources worthy of further research consideration. For instance, intervention studies to investigate whether training employees to become more mindfully and digitally confident helps to reduce negative well-being impacts in the digital workplace.

## Conclusion

As work is increasingly mediated by digital technology, organisations need to consider how to manage risks associated with the digital workplace alongside other psychosocial and physical risks in the workplace. Doing so effectively includes equipping employees with the right skills and tactics to protect well-being while working digitally, as well as considering wider technological and organisational-level solutions. Based on the evidence from this study, being mindfully and confidently digital should be considered important elements of living a healthy digital working life in the twenty-first century.

## Supporting information

**S1 Appendix. Construct definitions and key references.**
(DOCX)

**S2 Appendix. Measurement items.**
(DOCX)

**S3 Appendix. Figures for mediation analyses.**
(DOCX)

## Author Contributions

**Conceptualization:** Elizabeth Marsh, Elvira Perez Vallejos, Alexa Spence.

**Data curation:** Elizabeth Marsh.

**Formal analysis:** Elizabeth Marsh.

**Investigation:** Elizabeth Marsh.

**Methodology:** Elizabeth Marsh, Elvira Perez Vallejos, Alexa Spence.

**Project administration:** Elizabeth Marsh.

**Resources:** Elizabeth Marsh.

**Supervision:** Elvira Perez Vallejos, Alexa Spence.

**Validation:** Alexa Spence.

**Writing – original draft:** Elizabeth Marsh.

**Writing – review & editing:** Elvira Perez Vallejos, Alexa Spence.

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
