## [Decision Letter · Decision Letter 0]

21 Sep 2023

PONE-D-23-22747Mindfully and confidently digital: a mixed methods study on personal resources to mitigate the dark side of digital workingPLOS ONE

Dear Dr. Marsh,

Thank you for submitting your manuscript to PLOS ONE. After careful consideration, we feel that it has merit but does not fully meet PLOS ONE’s publication criteria as it currently stands. Therefore, we invite you to submit a revised version of the manuscript that addresses the points raised during the review process.

We look forward to receiving your revised manuscript.

Kind regards,

Bo Pu, Ph.D.

Academic Editor

PLOS ONE

2. We noted in your submission details that a portion of your manuscript may have been presented or published elsewhere. [Based on the quantitative dataset, we have developed another paper focused on the effects of information overload and Fear of Missing out on mental health and exhaustion (it is under consideration elsewhere, and is uploaded here for transparency). This separation of the findings was a deliberate effort to maintain focus and clarity in the current study; the information overload paper does not use any of the same analyses as the current study.] Please clarify whether this or publication was peer-reviewed and formally published. If this work was previously peer-reviewed and published, in the cover letter please provide the reason that this work does not constitute dual publication and should be included in the current manuscript.

Reviewers' comments:

Reviewer's Responses to Questions

**Comments to the Author**

1. Is the manuscript technically sound, and do the data support the conclusions?

Reviewer #1: Partly

Reviewer #2: Yes

Reviewer #3: Yes

2. Has the statistical analysis been performed appropriately and rigorously? 

Reviewer #1: Yes

Reviewer #2: Yes

Reviewer #3: Yes

3. Have the authors made all data underlying the findings in their manuscript fully available?

Reviewer #1: Yes

Reviewer #2: Yes

Reviewer #3: No

4. Is the manuscript presented in an intelligible fashion and written in standard English?

Reviewer #1: No

Reviewer #2: Yes

Reviewer #3: Yes

5. Review Comments to the Author

Reviewer #1: Comment 1: While references are important to establish the background and context of your study, this excerpt seems to heavily rely on citations. It might be helpful to briefly summarize the key findings of some of the references mentioned, particularly those that are central to your argument, to provide context for readers who might not be familiar with the cited studies.

Comment 2: The transition from discussing potential negative effects of the digital workplace to introducing the concept of mindfulness and digital workplace confidence could be smoother. You should clearly establish how mindfulness and confidence could potentially address the negative effects and why these factors are important.

Comment 3: While the study employs the Job Demands-Resources theory as a theoretical lens, it would be helpful to explicitly outline how digital workplace dark side effects fit into this framework and how they interact with the job demands and resources. Additionally, some of the hypotheses, particularly H1 and H2, could be more explicitly linked to the theoretical foundations to provide a stronger rationale for their formulation.

Comment 4: The control variables mentioned (age, gender, tenure, etc.) are indeed important to consider, but it would be beneficial to elaborate on why these particular variables were chosen and how they might influence the relationships under investigation. Additionally, discussing potential alternative explanations for the observed relationships could enhance the robustness of the findings.

Comment 5: For hypotheses involving mediation and moderation (H1, H3), it's crucial to provide details about the statistical methods used to test these relationships. This includes describing the analytical techniques and software employed, as well as the criteria for establishing mediation or moderation.

Comment 6: You've mentioned that two participants were removed due to an attention check failure. Provide a bit more detail about the nature of the attention check and how it was designed to ensure participant engagement and data quality.

Comment 7: While the inclusion of an open-ended question is a good idea, it would be helpful to explain how the responses to this question were utilized in the analysis or discussion. This can provide insight into how qualitative data supplements the quantitative findings.

Comment 8: You've mentioned that the interview questions were based on theoretical concepts and measurement instruments used in the survey. Elaborate on how these questions were formulated to capture participants' experiences, perceptions, and insights in a qualitative context. This helps ensure that the interview questions align with the qualitative nature of the study.

Comment 8: in discussion it would be beneficial to clarify whether TM and DWC are seen as potential outcomes of well-being or as protective factors. The study hints at their role as antecedents but doesn't explicitly discuss the direction of causality. Understanding this can help in interpreting the findings more accurately

Comment 9: While the study mentions qualitative findings, it could benefit from a more integrated discussion of how these qualitative insights complement or support the quantitative results. Providing specific examples or quotations from interviews would enhance the qualitative aspect of the study.

Comment 10: It would be helpful to discuss how the findings align or differ from existing literature on the relationship between personal resources and well-being in digital work environments. This would provide context for readers and help highlight the study's unique contributions.

Comment 11: The study acknowledges that the sample size for qualitative data limits generalizability. As such, it would be helpful to discuss the potential for transferability of qualitative insights to other contexts and settings, highlighting the need for larger-scale qualitative investigations to confirm these findings.

Comment 12: While the study highlights the Job Demands-Resources model and the personal resources of trait mindfulness (TM) and digital workplace confidence (DWC) as fruitful avenues for further research, it could provide more specific recommendations for the types of studies or interventions that could advance the field. This would offer guidance to researchers on where to focus their efforts.

Reviewer #2: Great work. It makes a very insightful reading . The analysis are very well done and the recommendation is apt. A good demonstration of the principles of research , methodology is goes well , keep it up on this good research .

Reviewer #3: The paper tries to present TM and DWC as indicators of burnout and health for workers.

The paper had shown that one or the other can be use as determinants but significant correlations are not fully achieved to support the different hypotheses presented.

The declaration in the study limitations, in my opinion, could be the main reasons for these.

First, a single point data collection may capture the status of the workers on that particular point in time BUT does not capture the working conditions they face that will lead to stress/burnout and negative effect on health thus, TM/DWC correlation on burnout/poor health is not significantly correlated.

Second, it is apparent that the younger generation are more "technologically savvy" than the older generations. The "technological gap" could play more contributions to the relationship the paper tries to present.

Third, one possible "other factors" that could contribute to the relationship the paper seek is the type of work itself. It is not presented whether the participants were from what type of work/job eg administrative, factory, retail shops, sales, etc. We know that technology play varying degrees in each type of job/work much more on the type of industry and to correlate across the board may result to "mixed results".

I think the paper has some significant observations and I suggest to narrow down the reach of the paper, instead of presenting a relationship across a wide scope, it can narrow down to one or a few segments/section.

6. PLOS authors have the option to publish the peer review history of their article (what does this mean?). If published, this will include your full peer review and any attached files.

Reviewer #1: No

Reviewer #2: **Yes: **Haitham Medhat Aboulilah

Reviewer #3: No

---

## [Author Response · Author response to Decision Letter 0]

27 Oct 2023

Journal requirements

We have reviewed the formatting templates you have highlighted and ensured that all elements of our manuscript meet these requirements, including captions for supporting information files.

Use of quantitative dataset

Utilizing the quantitative dataset, we have crafted an additional research paper that delves into the impacts of information overload and the Fear of Missing Out (FoMO) on mental health and exhaustion. This supplementary paper is currently under consideration by SAGE Open, and we have provided it with our submission for the sake of transparency. This deliberate division of findings was aimed at preserving the precision and clarity of the present paper. It's important to note that the paper on information overload employs entirely distinct analyses that would not be appropriate to include in the ‘Mindfully and digitally confident’ paper.

Repository information

The dataset has now been deposited on the UK Data Service’s Reshare site: https://reshare.ukdataservice.ac.uk/856732

 

Responses to Reviewers’ Comments

Reviewer #1: Comment 1: While references are important to establish the background and context of your study, this excerpt seems to heavily rely on citations. It might be helpful to briefly summarize the key findings of some of the references mentioned, particularly those that are central to your argument, to provide context for readers who might not be familiar with the cited studies.

Thank you for this comment. As a result, we have drawn out/summarized a number of key references further, see p4-8.

.

Comment 2: The transition from discussing potential negative effects of the digital workplace to introducing the concept of mindfulness and digital workplace confidence could be smoother. You should clearly establish how mindfulness and confidence could potentially address the negative effects and why these factors are important.

We have made some additions to the mindfulness and digital workplace confidence introductory sections (p7-9) and brought forward mention of them in the very first paragraph (p3).

Comment 3: While the study employs the Job Demands-Resources theory as a theoretical lens, it would be helpful to explicitly outline how digital workplace dark side effects fit into this framework and how they interact with the job demands and resources. Additionally, some of the hypotheses, particularly H1 and H2, could be more explicitly linked to the theoretical foundations to provide a stronger rationale for their formulation.

We have added information on how digital workplace dark side effects fit into the Job Demands-Resources model (p3-4), thank you for highlighting this omission. We have also tried to more clearly show how H1 and H2 are linked to the theoretical foundation of JD-R model (p6).

Comment 4: The control variables mentioned (age, gender, tenure, etc.) are indeed important to consider, but it would be beneficial to elaborate on why these particular variables were chosen and how they might influence the relationships under investigation. Additionally, discussing potential alternative explanations for the observed relationships could enhance the robustness of the findings.

Further information on the controls has been added in the sub-section on Measures, see p13.

Comment 5: For hypotheses involving mediation and moderation (H1, H3), it's crucial to provide details about the statistical methods used to test these relationships. This includes describing the analytical techniques and software employed, as well as the criteria for establishing mediation or moderation.

We have added further information about these analyses, see p18.

 

Comment 6: You've mentioned that two participants were removed due to an attention check failure. Provide a bit more detail about the nature of the attention check and how it was designed to ensure participant engagement and data quality.

Additional information about the attention check has been added, see p12.

Comment 7: While the inclusion of an open-ended question is a good idea, it would be helpful to explain how the responses to this question were utilized in the analysis or discussion. This can provide insight into how qualitative data supplements the quantitative findings.

We have clarified this point, see p12. 

Comment 8: You've mentioned that the interview questions were based on theoretical concepts and measurement instruments used in the survey. Elaborate on how these questions were formulated to capture participants' experiences, perceptions, and insights in a qualitative context. This helps ensure that the interview questions align with the qualitative nature of the study.

We have elaborated on this point, see p14.

Comment 8: in discussion it would be beneficial to clarify whether TM and DWC are seen as potential outcomes of well-being or as protective factors. The study hints at their role as antecedents but doesn't explicitly discuss the direction of causality. Understanding this can help in interpreting the findings more accurately.

We have further clarified the protective role of TM and DWC towards employee well-being in line with this comment, see p34.

Comment 9: While the study mentions qualitative findings, it could benefit from a more integrated discussion of how these qualitative insights complement or support the quantitative results. Providing specific examples or quotations from interviews would enhance the qualitative aspect of the study.

We have extended the qualitative commentary in the discussion and woven in some apposite quotes from the findings, e.g. p32, p34, p35.

Comment 10: It would be helpful to discuss how the findings align or differ from existing literature on the relationship between personal resources and well-being in digital work environments. This would provide context for readers and help highlight the study's unique contributions.

We have added commentary on this point, see p33, 34.

Comment 11: The study acknowledges that the sample size for qualitative data limits generalizability. As such, it would be helpful to discuss the potential for transferability of qualitative insights to other contexts and settings, highlighting the need for larger-scale qualitative investigations to confirm these findings.

We have added a comment on this in the limitations, see p35.

Comment 12: While the study highlights the Job Demands-Resources model and the personal resources of trait mindfulness (TM) and digital workplace confidence (DWC) as fruitful avenues for further research, it could provide more specific recommendations for the types of studies or interventions that could advance the field. This would offer guidance to researchers on where to focus their efforts.

We have elaborated this recommendation, see p35-6.

Reviewer #2: Great work. It makes a very insightful reading . The analysis are very well done and the recommendation is apt. A good demonstration of the principles of research , methodology is goes well , keep it up on this good research .

Thank you.

Reviewer #3: The paper tries to present TM and DWC as indicators of burnout and health for workers.

The paper had shown that one or the other can be use as determinants but significant correlations are not fully achieved to support the different hypotheses presented.

The declaration in the study limitations, in my opinion, could be the main reasons for these.

First, a single point data collection may capture the status of the workers on that particular point in time BUT does not capture the working conditions they face that will lead to stress/burnout and negative effect on health thus TM/DWC correlation on burnout/poor health is not significantly correlated.

Regarding capturing data at a single point in time, yes we agree that this is a limitation and have added comments on it in the Limitations section in response to your feedback, see p35.

Second, it is apparent that the younger generation are more "technologically savvy" than the older generations. The "technological gap" could play more contributions to the relationship the paper tries to present.

This is an interesting point. We have added some further commentary on the effects of age under Measures, see p13. However, in the present findings, age did not have a significant effect in any of the regressions, therefore we have not commented on it further in the results.

Third, one possible "other factors" that could contribute to the relationship the paper seek is the type of work itself. It is not presented whether the participants were from what type of work/job eg administrative, factory, retail shops, sales, etc. We know that technology play varying degrees in each type of job/work much more on the type of industry and to correlate across the board may result to "mixed results".

Thank you for flagging this up, we have added acknowledgement of this point in the limitations, see p35.

I think the paper has some significant observations and I suggest to narrow down the reach of the paper, instead of presenting a relationship across a wide scope, it can narrow down to one or a few segments/section.

We have indeed suggested that future researchers consider a narrower scope e.g. within a single organisation or team. Regarding the present findings, in responding to the various review comments we have aimed to further clarify them for readers.

---

## [Decision Letter · Decision Letter 1]

27 Nov 2023

Mindfully and confidently digital: a mixed methods study on personal resources to mitigate the dark side of digital working

PONE-D-23-22747R1

Dear Dr. Marsh,

We’re pleased to inform you that your manuscript has been judged scientifically suitable for publication and will be formally accepted for publication once it meets all outstanding technical requirements.

Kind regards,

Bo Pu, Ph.D.

Academic Editor

PLOS ONE

Additional Editor Comments (optional):

I suggest that this manuscript should be published.

Reviewers' comments:

Reviewer's Responses to Questions

**Comments to the Author**

1. If the authors have adequately addressed your comments raised in a previous round of review and you feel that this manuscript is now acceptable for publication, you may indicate that here to bypass the “Comments to the Author” section, enter your conflict of interest statement in the “Confidential to Editor” section, and submit your "Accept" recommendation.

Reviewer #1: All comments have been addressed

Reviewer #3: All comments have been addressed

2. Is the manuscript technically sound, and do the data support the conclusions?

Reviewer #1: Yes

Reviewer #3: Yes

3. Has the statistical analysis been performed appropriately and rigorously? 

Reviewer #1: Yes

Reviewer #3: Yes

4. Have the authors made all data underlying the findings in their manuscript fully available?

Reviewer #1: Yes

Reviewer #3: Yes

5. Is the manuscript presented in an intelligible fashion and written in standard English?

Reviewer #1: Yes

Reviewer #3: Yes

6. Review Comments to the Author

Reviewer #1: After carefully reviewing the comments, I have taken the necessary steps to address each point raised. I have thoroughly revised the paper based on their feedback, ensuring that all concerns have been adequately addressed.

Considering the comprehensive revisions made and the significant improvements achieved, I highly recommend accepting the paper for publication. The revisions have not only strengthened the overall quality of the research but have also enhanced its contribution to the field.

Reviewer #3: The necessary revisions were made to clarify several points which will make the manuscript achieved its objectives.

7. PLOS authors have the option to publish the peer review history of their article (what does this mean?). If published, this will include your full peer review and any attached files.

Reviewer #1: No

Reviewer #3: No

---

## [Editor Report · Acceptance letter]

1 Dec 2023

PONE-D-23-22747R1 

Mindfully and confidently digital: a mixed methods study on personal resources to mitigate the dark side of digital working 

Dear Dr. Marsh:

I'm pleased to inform you that your manuscript has been deemed suitable for publication in PLOS ONE. Congratulations! Your manuscript is now with our production department. 

Kind regards, 

on behalf of

Dr. Bo Pu 

Academic Editor

PLOS ONE